# A Novel Network Science and Similarity-Searching-Based Approach for Discovering Potential Tumor-Homing Peptides from Antimicrobials

**DOI:** 10.3390/antibiotics11030401

**Published:** 2022-03-17

**Authors:** Maylin Romero, Yovani Marrero-Ponce, Hortensia Rodríguez, Guillermin Agüero-Chapin, Agostinho Antunes, Longendri Aguilera-Mendoza, Felix Martinez-Rios

**Affiliations:** 1School of Chemical Sciences and Engineering, Yachay Tech University, Hda. San Jose s/n y Proyecto Yachay, Urcuqui 100119, Ecuador; maylin.romeroh@gmail.com (M.R.); hmrodriguez@yachaytech.edu.ec (H.R.); 2Universidad San Francisco de Quito (USFQ), Grupo de Medicina Molecular y Traslacional (MeM&T), Colegio de Ciencias de la Salud (COCSA), Escuela de Medicina, Edificio de Especialidades Médicas, Diego de Robles y vía Interoceánica, Pichincha, Quito 170157, Ecuador; 3CIIMAR/CIMAR, Centro Interdisciplinar de Investigação Marinha e Ambiental, Universidade do Porto, Terminal de Cruzeiros do Porto de Leixões, Av. General Norton de Matos, s/n, 4450-208 Porto, Portugal; gchapin@ciimar.up.pt (G.A.-C.); aantunes@ciimar.up.pt (A.A.); 4Departamento de Biologia, Faculdade de Ciências, Universidade do Porto, Rua do Campo Alegre, 4169-007 Porto, Portugal; 5Departamento de Ciencias de la Computación, Centro de Investigación Científica y de Educación Superior de Ensenada (CICESE), Ensenada 22860, Baja California, Mexico; longendri@gmail.com; 6Facultad de Ingeniería, Universidad Panamericana, Augusto Rodin No. 498, Insurgentes Mixcoac, Benito Juárez, Ciudad de México 03920, Mexico; felix.martinez@up.edu.mx

**Keywords:** cancer, tumor-homing peptide, in silico drug discovery, complex network, chemical space network, centrality measure, similarity searching, group fusion, motif discovery, starPep toolbox software

## Abstract

Peptide-based drugs are promising anticancer candidates due to their biocompatibility and low toxicity. In particular, tumor-homing peptides (THPs) have the ability to bind specifically to cancer cell receptors and tumor vasculature. Despite their potential to develop antitumor drugs, there are few available prediction tools to assist the discovery of new THPs. Two webservers based on machine learning models are currently active, the TumorHPD and the THPep, and more recently the SCMTHP. Herein, a novel method based on network science and similarity searching implemented in the starPep toolbox is presented for THP discovery. The approach leverages from exploring the structural space of THPs with Chemical Space Networks (CSNs) and from applying centrality measures to identify the most relevant and non-redundant THP sequences within the CSN. Such THPs were considered as queries (Qs) for multi-query similarity searches that apply a group fusion (MAX-SIM rule) model. The resulting multi-query similarity searching models (SSMs) were validated with three benchmarking datasets of THPs/non-THPs. The predictions achieved accuracies that ranged from 92.64 to 99.18% and Matthews Correlation Coefficients between 0.894–0.98, outperforming state-of-the-art predictors. The best model was applied to repurpose AMPs from the starPep database as THPs, which were subsequently optimized for the TH activity. Finally, 54 promising THP leads were discovered, and their sequences were analyzed to encounter novel motifs. These results demonstrate the potential of CSNs and multi-query similarity searching for the rapid and accurate identification of THPs.

## 1. Introduction

Cancer is a group of diseases developed in different cell and tissue types, and corresponds to the second leading cause of death globally [1]. It is based on the abnormal growth of cells due to an inherited genetic mutation or induced by the environment [2]. Despite novel therapy development for cancer treatment, improving chemotherapeutic drugs’ specificity towards cancer cells remains a challenge [2,3]. Additionally, cancer cells are generating multi-drug resistance (MDR) [4]. Consequently, in the pharmaceutical industry, there is a need to develop new anticancer agents with a different mode of action to tackle the current drug resistance of cancer cells without being cytotoxic to healthy ones [2]. To fill this gap, peptides have emerged as a potential therapeutic alternative against cancer. From 2015 to 2019, 15 peptides or peptide-containing molecules were approved by the FDA as drugs, demonstrating the growing interest of the scientific community [5].

Peptides have different biochemical and therapeutic properties than small molecules and proteins, making them attractive to the pharmaceutical and biotechnological industry [6,7]. Being smaller than proteins allows peptides to penetrate tissues more easily, have low cost, more accessible synthesis, and do not require folding to be biologically active [8]. In contrast to small molecules, they have a higher specificity and efficacy due to representing the smallest functional part of a protein [9]. Moreover, they are not supposed to interact with the immune system, are biocompatible, have tunable bioactivity, and have low cytotoxicity due to their degradation products being amino acids [10,11,12,13,14]. Hence, peptide-based drugs open a new door to an improved cancer diagnosis and treatment.

Tumor blood and lymphatic vasculature differ molecularly and morphologically from normal lymphatic and blood vessels [15]. Tumor-homing peptides (THPs) take advantage of this peculiarity. Thus, they are widely investigated as drug carriers and for imaging purposes on oncology treatments and diagnosis [16]. The first-generation of THPs have RGD (Arg-Gly-Asp) and NGR (Asn-Gly-Arg) motifs. RGD peptides have the characteristic of selectively binding to α integrins expressed in vascular endothelial cells of the tumor and metastatic tumor cells, and NGR to aminopeptidase N (APN) receptors [17,18]. Although, there are neither non-RGD nor NGR peptides that home tumor blood vasculature and cancer cells by interactions with other receptors, such as the endothelial growth factor receptor (EGFR) [19,20,21,22,23].

THPs are discovered by using in vitro and ex vivo/in vivo phage display technology, which is time-consuming, expensive, and may not translate to humans due to differences between the animal models and humans [24,25,26]. For these reasons, bioinformatics tools such as databases and webservers are being employed for the accurate prediction of novel THPs [26,27,28]. In this way, short sets of the most promising THPs become the candidates for posterior experimental verification.

To date, the databases available for experimentally validated THPs are TumorHoPe (includes 744 THPs) [27] and starPepDB (includes 659 THPs) [29], and the available TH activity predictors are TumorHPD (https://webs.iiitd.edu.in/raghava/tumorhpd) (accessed on 1 May 2021) [26], THPep (http://codes.bio/thpep) (accessed on 1 May 2021) [28], and SCMTHP (SCMTHP (pmlabstack.pythonanywhere.com) (accessed on 5 January 2022) [30]. TumorHPD uses the supervised ML method Support Vector Machine (SVM) as a classifier with three features: amino acid composition, dipeptide composition, and binary profile patterns, achieving 86.56% as the highest accuracy [26]. The second ML method, THPep, has a Random Forest (RF) classifier with three features: amino acid composition, dipeptide composition, and pseudo amino acid composition, resulting in 90.13% of maximum overall accuracy [28]. However, the datasets used for training and testing both ML models contain peptides with highly similar sequences. On the other hand, SCMTHP is the most recently reported method based on the scoring card method (SCM) [30]. It determines the propensity scores for the amino acids’ and dipeptides’ composition accounting for THP sequences and applies a threshold value to discriminate between THP and non-THPs. Nonetheless, the performance of SCMTHP is similar to ML-based predictors, achieving a maximum accuracy of 82.7%.

Recently, Marrero-Ponce et al. published a new software named starPep toolbox (http://mobiosd-hub.com/starpep/) (accessed on 2 February 2021), which is aimed to perform network analyses on the integrated graph database called starPepDB, which include the most comprehensive and non-redundant database of antimicrobial peptides (AMPs) [29,31]. Here, we propose an alternative methodology to identify potential THPs by combining network science with multi-query similarity searching against the AMPs of starPepDB. We used the starPep toolbox software as the main bioinformatics tool and the Chemical Space Network (CSN) to represent the chemical space of peptides as a coordinate-free system. To the best of our knowledge, there are no reported studies where data mining and screening is supported by network science to discover peptides for pharmaceutical purposes [29]. Firstly, we built models of representative and non-redundant THPs using centrality analysis and supervised retrospective similarity searching to perform the TH activity prediction. The outstanding model, named THP1, predicted the TH activity of three benchmarking datasets of THPs/non-THPs achieving accuracies between 92.64–99.18% and Matthews Correlation Coefficient (MCC) between 0.894–0.98, demonstrating the feasibility of this new methodology. Then, we performed a hierarchical screening for drug repurposing using network-based algorithms implemented in the starPep toolbox, the best model THP1, local alignments, and webservers to predict relevant activities related to the TH. Their TH activity was optimized by generating random libraries, where the peptide undergoes amino acid’s stochastic substitutions at different positions. Finally, a set of 54 potential THPs from AMPs was proposed, where common motifs were identified.

## 2. Materials and Methods

The overall workflow of this report, shown in Figure 1, was based on two steps: (i) generation/selection of the model of representative THPs from starPepDB in starPep toolbox, and (ii) prediction of potential new THPs from AMPs. In the first step, some models of representative THPs from starPepDB were built using different centrality measures to rank the nodes and extract the representative and less similar sequences by applying local alignment. Then, the best multi-query similarity searching model (SSM) was selected by the classification performance and its ability to correctly retrieve THPs from benchmark THPs databases by using group fusion (MAX-SIM rule) similarity searching. 

In the second step, the model was used to perform similarity searching to repurpose AMPs as THPs from starPepDB, and their TH activity was optimized using the TumorHPD server. Additionally, sequence motifs were found from the set of potential THPs using multiple sequence alignments [32,33,34,35], alignment-free methods [36], and PROSITE server (https://www.genome.jp/tools/motif) (accessed on 15 July 2021).

### 2.1. StarPep Toolbox Software

The starPep toolbox uses FASTA files as inputs and includes the starPepDB. Peptides are represented as nodes connected by an edge if they have any relationship. It can perform querying, filtering, visualization of networks, scaffold extractions, single or multiple queries similarity searching, and analysis of peptides by graph networks [29,31].

Networks can be built based on the metadata of peptides or based on the pairwise similarity measures calculated for their respective sequence. In metadata networks, nodes are connected by a specific parameter in common, such as origin; the target against which they are assessed; functionality; the database where they come from; the cross-reference; N-terminus; C-terminus; or amino acid composition. In similarity networks, peptides are codified by descriptors, such as length, net charge, isoelectric point, molecular weight, Boman index, indices based on aggregation operators, hydrophobic moment, average hydrophilicity, hydrophobic periodicity, aliphatic index, and instability index [29,31,37]. Moreover, networks are visualized using different layouts, such as Fruchterman–Reingold [38].

Networks can be clustered, and communities are optimized using the Louvain method [39]. Moreover, the centrality of each node can be particularly measured by harmonic, community hub-bridge, betweenness, and weighted degree. Centrality is crucial to perform scaffold extractions because peptides are ranked according to their centrality score, and then redundant sequences are removed, prioritizing the most central. Thus, scaffold extractions depend on the type of centrality applied.

On the other hand, similarity searching, which is the basis of this study, is performed using a set of queries against a target dataset, where different percentages of identity (or similarity thresholds) can be applied. An identity score is a number between 0–1, and it is calculated using the Smith–Waterman local alignment with BLOSUM 62 substitution matrix [40]. Multiple queries similarity searching works using the group fusion model explained in the following section.

### 2.2. Model Selection

The dataset of reported THPs was extracted from starPepDB in the starPep toolbox. All 45120 peptides contained in starPepDB were filtered by the “Tumor Homing” query in the metadata function, where 659 entries were obtained (SI1-A).

#### 2.2.1. Network Analysis

##### Similarity Threshold Analysis

Network analysis of peptides was performed by building the CSN of 659 THPs in the starPep toolbox. To choose the appropriate similarity threshold to build the network of THPs, CSNs were built by varying in 0.05 the cut-off value from 0.10 to 0.90 (17 CSNs in total). Some metrics were retrieved from each CSN using the starPep toolbox, such as density, number of communities, modularity, and number of singletons.

By default, when CSN was built, nodes with higher than 98% of similarity were removed using the local alignment Smith–Waterman algorithm. The similarity metric used to establish the pairwise similarity relationships between nodes was the min–max normalized Euclidean distance. Then, a centrality was calculated and those nodes with 0 as vertex degree were identified as outliers and then removed, leaving the giant (or connected) components of the CSN, i.e., subgraph where all nodes are connected. In this case, community hub-bridge centrality was calculated. However, any centrality measure could have been calculated since singletons always have zero centrality. After that, the network was clustered and the modularity optimized using the modularity optimization algorithm based on the Louvain method [39].

The network was saved as a Graph ML file to be opened in Gephi [41] for subsequent calculation of ACC. Finally, density, modularity, and ACC as a function of similarity threshold were graphed in Origin to decide what similarity threshold is the best.

##### Network Characterization

CSN of the giant components derived from the application of the best similarity threshold was characterized by the number of nodes, edges, outliers, density, number of communities, and modularity. These parameters were obtained from starPep toolbox while ACC, diameter (larger shortest path), average path length, and a total of triangles were drawn from Gephi. These parameters allow knowing the topology and structural patterns of the CSN.

For network visualization, Force Atlas 2 was used as a layout algorithm where colors represent different clusters, and node size means how central the node is according to the community hub-bridge centrality. Network visualization aims to obtain an aesthetically pleasing and understandable graph where nodes are not overlapped.

On the other hand, CSN of outliers was built with a cut-off of 0.30 to procure an appropriate density; then, it was clustered. Moreover, a subsequent scaffold extraction was applied based on hub-bridge centrality, and on 30% identity from local alignment.

The network of outliers was characterized according to the number of nodes, edges, communities, density, modularity, average degree, ACC, diameter obtained before scaffold extraction, and the number of nodes and edges obtained after scaffold extraction. For network visualization, Fruchterman–Reingold was used as a layout algorithm; colors represent different clusters while node size displays how central it is according to hub-bridge measure.

#### 2.2.2. Centrality Analysis

The most influential nodes were used to find the new potential THPs, and centrality is the crucial parameter that provides this information. Thus, the four available centrality types in the starPep toolbox (weighted degree, community hub-bridge, betweenness, and harmonic) were calculated and normalized using the min–max method. Then, redundant peptides were removed by applying the scaffold extraction procedure that is described as follows: peptides were ranked based on the scores obtained after centrality calculation and we used 30% similarity cut-off of local identity from the Smith–Waterman algorithm to retrieve sets of sequences with a maximum of 30% similarity [40]. Subsequently, nodes with 10% lower centrality than the most central node were removed in each metric. The sets obtained after applying this process were named as 30 + 10%.

On the other hand, harmonic and weighted degree were calculated and normalized, and redundant peptides were removed by applying the scaffold extraction procedure using four different similarity cut-offs of local identity: 30, 40, 50, and 60%.

#### 2.2.3. Similarity Searching Model for THPs Prediction

This study’s proposed method for discovering potential THPs was based on similarity searching. For that reason, multiple query similarity searching models (SSMs) composed of several queries representing the most important and less redundant nodes of CSN and a similarity threshold were tested against datasets that contain well-known THPs/non-THPs through similarity searching. The recoveries from the similarity searching were statistically evaluated to select the best model for identifying potential THPs within the AMPs.

##### Query Datasets (Reference Sequences)

The retrieved sets after applying scaffold extractions at each centrality measure; the two sets of outliers; combinations of outliers with sets obtained from centrality-based scaffold extractions; and combinations between sets obtained from scaffold extractions performed using different centrality metrics were used as queries (Qs). In total, we tested 22 sets of Qs, where twelve sets resulted from the application of the scaffold extraction procedures as well as two sets of outliers, and eight sets resulted from the combination between sets.

##### Target Databases

Three training datasets that consider well-known THPs and randomly generated non-THPs [42] were used as the target or calibration for the recovery. THPep, TumorHPD, and SCMTHP employed these datasets for training their methods [26,30,42].

Main dataset: 651 experimentally validated THPs and 651 random non-THPs (SI1-B). They were collected from TumorHoPe [27] and the literature [26].Small dataset: 469 experimentally validated THPs and 469 random non-THPs (SI1-C). They are peptides derived from the Main dataset with 4 to 10 aa residues.Main90 dataset: 176 THPs and 443 non-THPs (SI1-D). They are peptides from the Main dataset with equal or lower than 90% of sequence similarity.Main and Small datasets were retrieved from Ref. [26], while Main90 from Ref. [27].

##### Group fusion

Group fusion is based on the variation in the query (reference peptide), but keeping constant the identity measure [43]. Each peptide’s identity score is calculated from the target dataset varying the Qs. The fusion group’s algorithm associates a fused score to each target peptide, i.e., the maximum similarity (MAX-SIM) scores from all resulting identity scores against the Qs. Therefore, considering peptide S from the target dataset, reference peptide Q from the Qs, the identity score I(S,Q), and the MAX-SIM score obtained, the algorithm assigns I(S,Q) as the fused score to peptide S. The local identities were calculated with the Smith–Waterman, and is a number between 0–1, with 1 being the maximum identity. The procedure is illustrated in Figure 2.

##### Retrospective Similarity Searching

Main Dataset was imported to starPep toolbox. The similarity searching was performed using the “Multiple query sequences” option of the software and the Qs obtained from 30 + 10% similarity cut-offs of local alignment and outliers. The group fusion is applied by default during the similarity searching, and results were ranked according to the fused score (MAX-SIM value). Subsequently, seven different percentages of identity (similarity thresholds), 30, 40, 50, 60, 70, 80, and 90%, were tested, where peptides with identities equal to or higher than the applied threshold were retrieved as predicted THPs. Figure 2 illustrates how the similarity searching works.

The rescued nodes, i.e., predicted THPs, were statistically evaluated to validate the prediction. Thus, it is possible to identify the two centrality measures and percentages of sequence identity with the best performance.

Then, similarity searching was performed using only the sets of the best two centrality measures as Qs: harmonic and weighted degree, and 30, 40, 50, 60, and 70% of identity. In Small and Main90 datasets, only the sets of harmonic and weighted degrees were used as Qs, applying 40, 50, and 60% of identity for recovery. In total, 98 different SSMs were evaluated.

#### 2.2.4. Statistical Analysis

The ability of the SSMs to predict THPs was validated by the measurement of their accuracy (Ac), kappa (κ), sensitivity (Sn), specificity (Sp), the precision of positives and negatives (P_pos_ and P_neg_, respectively), MCC, and false accept rate (FAR%) using the following formulas.
(1)$$\mathrm{Ac}=\frac{\mathrm{TP}+\mathrm{TN}}{\mathrm{TP}+\mathrm{TN}+\mathrm{FP}+\mathrm{FN}},$$
(2)$$\kappa =\frac{\mathrm{Po}-\mathrm{Pc}}{1-\mathrm{Pc}},$$
(3)$$\mathrm{Sn}=\frac{\mathrm{TP}}{\mathrm{TP}+\mathrm{FN}}\;,$$
(4)$$\mathrm{Sp}=\frac{\mathrm{TN}}{\mathrm{TN}+\mathrm{FP}},$$
(5)$$P_{\mathrm{pos}}=\frac{\mathrm{TP}}{\mathrm{TP}+\mathrm{FP}}\;,$$
(6)$$P_{\mathrm{neg}}=\frac{\mathrm{TN}}{\mathrm{TN}+\mathrm{FN}}\;,$$
(7)$$\mathrm{MCC}=\frac{\mathrm{TP}\times \mathrm{TN}-\mathrm{FP}\times \mathrm{FN}}{\sqrt{\left(\mathrm{TP}+\mathrm{FP}\right)\times \left(\mathrm{TP}+\mathrm{FN}\right)\times \left(\mathrm{TN}+\mathrm{FP}\right)\times \left(\mathrm{TN}+\mathrm{FN}\right)}}\;,$$
(8)$$\mathrm{FAR}\%=\frac{\mathrm{FP}}{\mathrm{FP}+\mathrm{TN}}\times 100\;,$$
where, TP is the number of true positives, TN is the number of true negatives, FP is the number of false positives, FN is the number of false negatives, Po is the relative observed agreement between the observers equal to the Ac, and Pc is the expected chance agreement calculated by the formula $\mathrm{Pc}=\frac{\left(\mathrm{TP}+\mathrm{FP}\right)\times \left(\mathrm{TP}+\mathrm{FN}\right)+\left(\mathrm{FN}+\mathrm{TN}\right)\times \left(\mathrm{FP}+\mathrm{TN}\right)}{{\left(\mathrm{TP}+\mathrm{TN}+\mathrm{FP}+\mathrm{FN}\right)}^{2}}$.

Finally, the best 9 SSMs were compared and ranked using the Friedman test-based analysis performed in KEEL [44]. The Friedman test identified the best model based on the statistical metrics previously shown [45]. Moreover, it allowed us to compare the models and determine if their difference was statistically significant and not due to chance. The confusion or classification matrix of the best model was constructed. The best models were compared with reported ML models used for THP prediction, TumorHPD, and THPep, using the same three calibration datasets.

### 2.3. Identification of Potential THPs

#### 2.3.1. Hierarchical Screening

Drug repurposing is an alternative methodology widely applied to discover drugs because it reduces approval time for their clinical use [46,47]. Thus, firstly, we repurposed AMPs from starPepDB as THPs.

Pipeline Prospective Screening. First, AMPs without reported TH activity and toxicity with a sequence length between 3 and 25 residues were filtered from the chemical space of starPepDB. Secondly, the “Scaffold extraction” option removed AMPs with higher than 95% sequence similarity by local alignment. Thirdly, multiple query similarity searching was performed using the best SSM (THP1), obtained in the previous section, to explore the chemical space of non-THPs, non-toxic, and non-redundant peptides with a length of 3–25 aa, using 60% as similarity threshold. In the recovered set, peptides with a similarity score of 1 were removed.Activity Prediction. Peptides with reported tumor-homing activity in the literature were removed since the main objective of this study was to identify novel THPs. Then, theoretical activities of virtual hits were predicted using webservers TumorHPD [26], THPep [28], AntiCP [48], CellPPD [49], ToxinPred [50], and HemoPI [51], to corroborate their potential as THPs and prioritize those that do not harm healthy cells. The activities of interest were tumor homing, anticancer, cell-penetrating, toxicity, and hemolysis. The SVM thresholds used were 0.30 in servers TumorHPD, AntiCP, and CellPPD, and 0 in server ToxinPred.Redundancy Reduction by Network Analysis. CSN of hits was built, clustered, and the modularity was optimized using the Louvain method in the starPep toolbox. Then, harmonic and weighted degree centralities were calculated to perform a scaffold extraction using a 60% identity as the threshold.Visual Mining. The neighborhood of well-known THPs of each potential THP was visualized using the starPep toolbox. CSN of 659 THPs in starPepDB was built using 0.60 as cut-off, clustered, and optimized modularity. Hits obtained in the previous step after scaffold extraction were embedded into the CSN of 659 THPs to study the neighborhood of each peptide. Hence, the 3 nearest neighbors from 659 THPs directly attached to each hit were visualized. If 2 peptides shared the same 2 or 3-nearest neighbors, one of them was prioritized, choosing the one with better predicted activities.

#### 2.3.2. Tumor-Homing Activity Optimization

Lead hits detected from hierarchical virtual screening were AMPs from starPepDB with a natural or designed activity different from tumor homing. That is the reason why their tumor-homing action should be enhanced. Lead hits were optimized by punctual amino acid mutations using the “Designing of Tumor Homing Peptides” module of TumorHPD (https://webs.iiitd.edu.in/raghava/tumorhpd/peptide.php) (accessed on 10 September 2021), and the procedure is shown in Figure 3. Both lead and mutated sequences were shortened into fragments of 5, 10, and 15 residues in length using the same server.

The optimized sequences showing a higher tumor-homing activity score than parent hits were analyzed by CSN in the starPep toolbox using 0.60 as the similarity threshold to build the network. In addition, tumor homing, toxicity, hemolytic, anticancer, and cell penetrability were predicted using servers listed below: THPep (http://codes.bio/thpep), TumorHPD (https://webs.iiitd.edu.in/raghava/tumorhpd) (accessed on 25 September 2021), AntiCP (https://webs.iiitd.edu.in/raghava/anticp2) (accessed on 25 September 2021), CellPPD (https://webs.iiitd.edu.in/raghava/cppsite1) (accessed on 25 September 2021), ToxinPred https://webs.iiitd.edu.in/raghava/toxinpred (accessed on 25 September 2021), and HemoPI https://webs.iiitd.edu.in/raghava/hemopi (accessed on 25 September 2021). Redundant sequences with higher than 50% similarity were removed by scaffold extraction.

The optimized sequences and parent hits were merged, and the corresponding CSN was built using 0.50 of cut-off and clustered. Next, harmonic centrality was calculated. Each cluster was analyzed separately to prioritize the most central, potent, non-toxic, and non-hemolytic lead THPs. Finally, the heat map and histogram of pairwise sequence identity of lead compounds were constructed to explore their structural diversity.

#### 2.3.3. Motif Discovery

##### Multiple Sequence Alignments 

As the resulting potential THPs were hard-to-align sequences because of their short length and variability, they were grouped into seven clusters according to the neighborhood in the CSN. Given that two peptides underrepresented clusters 1 and 5, they were fused in a cluster labeled 1–5. Thus, peptide clusters (2–4, 1–5, and singletons) were aligned independently using multiple sequence alignments (MSA), publicly available at https://www.ebi.ac.uk/Tools/msa/ (accessed on 28 September 2021). Four different MSA algorithms were applied with their default parameters to determine consensus motifs within each cluster: (1) Clustal-Omega v 1.2.4 [32], (2) MAFFT (Multiple Alignment using Fast Fourier Transform) v7.487 with the iterative refinement FFT-NS-i option [33], (3) MUSCLE (Multiple Sequence Comparison by Log-Expectation) v3.8 [34], and T-Coffee (Tree-based Consistency Objective Function for Alignment Evaluation) v1.83 [35].

The resulting MSAs were employed to extract the conserved motifs by considering the consensus sequences estimation from the programs Jalview v2.11.1.4 [52], EMBOSS Cons v6.6.0 (https://www.ebi.ac.uk/Tools/msa/emboss_cons/) (accessed on 28 September 2021), and Seq2Logov2.1 (http://www.cbs.dtu.dk/biotools/Seq2Logo/) (accessed on 28 September 2021) [53].

##### Alignment-Free Method

Peptides were analyzed in STREME [36] (Sensitive, Thorough, Rapid, Enriched Motif Elicitation) to discover fixed-length patterns (ungapped motifs) that were enriched with respect to a control set generated by shuffling input peptides [52]. The analyses were performed via its webserver (https://meme-suite.org/meme/tools/streme) (accessed on 28 September 2021), by considering both total peptides and by each cluster. The motif width was set between 3–5 amino acids length. STREME applies a statistical test at *p*-value threshold = 0.05 to determine the enrichment of motifs in the input peptides compared to the control set.

##### Motif Search in PROSITE

Peptides were queried by the Motif Search tool (https://www.genome.jp/tools/motif/) (accessed on 28 September 2021) and integrated into the GenomeNet Suite (https://www.genome.jp/) (accessed on 28 September 2021). PROSITE Pattern and PROSITE Profile libraries were only considered for the motif search.

## 3. Results and Discussion

### 3.1. Model Selection

#### 3.1.1. Network Analysis

##### Similarity Threshold Analysis

Out of the set of 659 THPs retrieved from starPepDB, 627 peptides (SI1-A-I) were filtered with lower than 98% similarity by local alignment. The adequate similarity threshold was chosen before building CSN with the 627 peptides. This step is non-trivial since it is the parameter that defines the topology and network features [54]. Hence, the appropriate cut-off for building the CSN was determined based on the variability of network parameters such as density, modularity, ACC, and singletons at different cut-off similarity values. Graphml files corresponding to the 17 CSNs are available at SI2. Appendix A shows the obtained parameters at each cut-off.

The graph of density, modularity, and ACC as a function of the similarity threshold is shown in Figure 4. The density is lower at a higher similarity threshold. ACC follows the same pattern until the 0.65 similarity threshold. By contrast, modularity increases as the similarity threshold increases, while the clustering is optimized.

A well-defined network needs a compromise among the density, modularity, and ACC parameters, but also accounts for the number of outlier nodes because they are atypical peptides with particular properties. Networks with very low density display too many outliers (see Appendix A), while networks with very high density show a massive connection. In both cases, information is lost and interpretation becomes difficult. According to the literature, the best density percentages are generally around 1% or 2.5% because they generate high modularity but allow an adequate understanding of the network [54]. As modularity indicates the existence of community structures, the ideal value must show an equilibrium between a non-clustered network and an artificially clustered network due to the high modularity value. In this sense, the selected similarity threshold was 0.60, where CSN shows the best trade-off among network parameters and connectivity: 2.3% of density, 0.47 of modularity, 0.428 of ACC, and 99 outliers (15.8% of overall nodes). Therefore, the giant components of the network were 528 nodes (SI1-A-II).

##### Network Characterization

Some parameters such as density, number of clusters, modularity, average degree, ACC, and diameter were calculated and shown in Table 1 to get an overview on the giant component and outliers of the CSNs, which are represented in Figure 5 and Figure 6, respectively.

Additionally, the degree of distribution of the giant components is shown in Figure 7. It gives some information about the structure of the CSN. In this case, the distribution degree is concentrated in the nodes with low vertex degrees. However, it has a tail associated with the nodes with higher vertex degrees in a lower proportion. The nodes with higher degrees correspond to the most central nodes, which, as can be corroborated in Figure 5, are in the minority.

Outliers are relevant THPs because they present characteristics regarding 528 nodes that make up the giant component; so, they are unique or atypical sequences. CSN of the 99 singletons (SI1-E) was built using 0.30 of similarity threshold (Figure 6a). Then, sequences with higher similarity than 30% by local alignment were removed based on hub-bridge centrality ranking, where 34 outliers (SI1-E-I) with unique sequences were obtained (Figure 6b).

#### 3.1.2. Centrality Analysis and Similarity Searching

Centrality is the crucial parameter to build the model that will be proposed to identify THPs. It allows the identification of the most influential sequences of the giant components. SI3 (Excel file) shows the normalized centrality measurements of 528 THPs. On the other hand, outliers are nodes with unique properties that enrich the influential sequences model. Therefore, both sets from centrality measurements and sets of outliers represent the chemical space of THPs and will be used as queries to perform the similarity searching against the target datasets. In total, 98 different SSMs were generated based on 22 query sets (FASTA files available at SI4) and similarity thresholds between 0.3 and 0.9.

The predictions and performance of the 98 SSMs are shown in SI5 and SI6-A, respectively, where active and inactive labels indicate predicted THPs and non-THPs, respectively. In general, it is observed that the performance of query datasets followed the following tendency of relevance: weighted degree > harmonic > hub-bridge > betweenness > singletons (outliers). However, the combination of query datasets from different centrality types overperforms the sets selected with only one centrality measure. The addition of the outliers set improved the performance of the combination sets since it generates the complete representation of the chemical space of THPs. Moreover, better performance was obtained using 40, 50, and 60% identity in the similarity searching.

The performance of the best nine SSMs to predict activity in Main, Small, and Main90 datasets are shown in Table 2, Table 3, and Table 4, respectively, where H is the set obtained when harmonic centrality was calculated, W is the set obtained when the weighted degree was calculated, and sing is the set of 99 outliers.

It can be noticed that the best statistics were achieved using the query composed of the union of harmonic and weighted degree, both using 60% similarity cut-off of local alignment during scaffold extraction, and the 99 outliers sets, comprising in total 479 query sequences. Moreover, 60% was the best percentage of identity where there was a compromise for all statistical parameters. All statistical parameters showed values higher than 0.88.

The best nine SSMs were compared and ranked using the Friedman test by comparing multiple statistical metrics from each SSM on the three target datasets (details in SI6-B). The best SSM was the set **CSN-TH-0.60Sc-479-H+W+s-0.6-583 (479Q_0.6)**, named THP1, showing excellent statistical metrics (>0.85) for the model (shown in Table 2, Table 3 and Table 4). It is composed of the union of nodes with an identity lower than 60% from the global centrality harmonic with those obtained from applying weighted degree and the set of 99 outliers (479 nodes). The best percentage of identity used to search similarity was 60%. The confusion matrices of THP1 are shown in SI6-C. It can be seen that the prediction of the model was not at random as the MCC was much greater than zero [55].

Finally, the Friedman test of the THP1 versus the reported models used in TumorHPD [26] and THPep [28] servers revealed there is a significant difference between the models, being that the performance of the similarity searching methodology is superior (details in SI6-C and SI6-D). Figure 8 shows the ranking scores from the test, where THP1 is the first ranked method. Finally, Table 5 compares between the model on the three benchmarking datasets. The MCC of predictions using THP1 improved by an average of 28.76% over ML-based models.

### 3.2. Identification of Potential THPs

#### 3.2.1. Hierarchical Screening

Starting from the 45120 AMPs contained in starPepDB, and after applying the previously explained filters and performing the similarity searching, 43 lead hits were retrieved (SI7-A). Figure 9 shows the step-by-step hierarchical virtual screening. Until today, these repurposed sequences have not reported any tumor-homing activity.

#### 3.2.2. Tumor-Homing Activity Optimization

A library of 180 sequences (SI7-B) was obtained from the optimization of 43 hits in TumorHPD. They have a higher TH score, lower toxicity, and hemolytic activity than the originals. Mutations enriched the sequences with W and C residues. Mainly G and V residues from the originals were mutated to W, while R and K were to C. Studies report the presence of W contributes positively to the intracellular translocation of peptides [56]. Additionally, it was reported that W enhances the stability of peptides in serum and salt [57].

Forty-one peptides from the library were prioritized by studying their CSN, where 50% scaffold extraction by local alignment was accomplished. The sequences were clustered and ranked according to the global harmonic centrality to perform the scaffold extraction. Only the most central sequences with a similarity among them lower than 50% were kept. Forty-one sequences have higher predicted TH activity by TumorHPD than the original peptides, with scores between 0.39 and 1.92. Furthermore, they are anticancer and have less toxicity and hemolytic activity. 12 out of 41 sequences come from fragments of original sequences of 5, 10, and 15 lengths; 15 resulted from four punctual mutations from the originals; and 14 from fragments of mutated sequences of 5, 10, and 15 lengths. Two out of forty-one peptides, CNGRCGGKLA and WCAMS, are part of reported THPs, validating the novel methodology to discover potential THPs. CNGRCGGKLA is the *N*-end of the CNGRCGGKLAKLAKKLAKLAK peptide containing the NGR TH motif and a disulfide bridge that gives stability. CNGRCGGKLAKLAKKLAKLAK binds to CD13 of tumor cells acting as ACP and THP [58]. At the same time, WCAMS is the *C*-end of the KLWCAMS peptide that homes mouse B16B15b melanoma [59].

We selected the most promising 13 sequences from the 43 lead hits and these were combined with the 41 optimized hits. In total, we proposed 54 peptides (SET 1, FASTA file in SI7-C) with a diverse molecular structure, low toxicity, and hemolytic activity, with most of them also showing potential anticancer activity (SI7-D). Among the 54 lead hits, only one sequence has the well-known NGR motif. Therefore, SET 1 is composed of new structural entities within the known structural space of the THPs.

The sequence diversity of SET 1 was evaluated using all vs. all global alignment where pairwise sequence identities were explored. As shown in Figure 10, the 54 lead hits present a structure singularity sharing pairwise identities with 30%.

#### 3.2.3. Motif Discovery

As a consequence of the structural diversity of SET 1, the discovery of motifs accounting for the TH activity is not a straightforward task. In this sense, sensitive multiple sequence alignment (MSA) tools and alignment-free (AF) approaches (e.g., STREME) were applied to unravel new TH motifs.

The resulting 54 lead THPs were mapped onto CSN space to identify putative communities and make possible the application of MSA algorithms for motif identification. These networks communities were considered clusters containing related peptides. Finally, six clusters were conformed with 14, 10, 8, 4, 10, and 8 members, respectively (SI7-E). The last cluster grouped the singletons (peptides identified as atypical in the CSN).

Clustal-Omega [32], MAFFT [33], MUSCLE [34], and T-Coffee [35], which are MSA algorithms developed after the classical ClustalW, were applied, so that they can deal with hard-to-align sequences shown in each cluster, and thus to detect any conserved signature or motif. Since each MSA has implemented a different algorithm to improve alignment quality, their consideration for the estimation of consensus regions helped us identify TH motifs by using the Jalview, EMBOSS Cons and Seq2Logo programs (SI8). As the EMBOSS Cons, gives a more legible output, only displaying high scored amino acids/positions (capital letters), less scored but positive residues (lower-case letters), and non-consensus positions (x), were selected as the primary source to set consensus/conserved regions. The non-consensus positions were estimated using default parameters by visual inspection of the corresponding positions in the Jalview program [52] and the Seq2Logo [53]. Table 6 depicts the consensus motifs, unraveled by each MSA algorithm.

None of the motifs found by MSA have been reported as TH motifs (Table 6). However, one of the motifs from No. 3 CxxxR, CGGCR, contains the CXXC motif, which is the active site of thioredoxin (Trx), a relevant protein in mammalian cells that acts as an antioxidant and participates in programmed cell death inhibition and cell growth, commonly used as a target for cancer treatments [60,61]. Moreover, CWKG (No. 5) is contained in a nanoscale molecular platform used as a drug delivery system in chemotherapy to enhance the conjugation of mitomycin C to the carrier [62].

On the other hand, the AF approach STREME was used to find unaligned patterns ranging from 3–5 aa length within the overall 54 peptides and each peptide cluster. STREME has been recently reported as the most accurate and sensitive algorithm among its competing state-of-the-art partners [36]. Unlike previous algorithms [63,64,65], STREME uses a position weight matrix (PWM) to count position matches efficiently for a motif candidate against a Markov model built up from shuffling the same input set (control sequences). Table 7 displays the enriched motifs, discriminating the 54 lead peptides against the control sequences. The same search was also performed by considering each cluster content. Motifs appearing in more than 20% of the query sequences are listed according to their statistical significance (score).

One of the motifs discovered by STREME had been reported as tumor-homing, WRP interacting with VEGF-C [66,67]. Other found motifs have been reported but not as TH, such as WRPW, PRW, WKG, and PSHL. WRPW is the binding site of the 7 Enhancer of split E(spl) basic helix–loop–helix (bHLH) protein and the hairy protein to the corepressor protein Groucho-TLE via WD40 domain [68]. PRW is part of a biocatalyst, which is conjugated to a lipid by an ester or amide bond [69]. WKG is a ribosomally synthesized and post-translationally modified peptide [70] and PSHL is a tetrapeptide that affects HIV-1 protease (PR) [71].

Lastly, 54 lead THPs were queried against PROSITE’s pattern and profile databases using the search engine Motif Search of the GenomeNet suite [72]. Only two query peptides, which are shown in Table 8, had significant matches with motifs found in gonadotropin-releasing hormones (GnRH) and bombesin-like peptides. 

These two peptide signatures and their receptors are involved in neuroendocrine signaling pathways associated with physiological states and tumors. GnRH is the hypothalamic decapeptide that plays a crucial role controlling women’s reproductive cycle. GnRH binds to specific receptors on the pituitary gonadotrophic cells, but it also is expressed in other reproductive organs, e.g., ovaries, and tumors derived from the ovaries. It has been shown GnRH is involved in ovarian cancer regulation proliferation and metastasis either by the indirect signaling pathway or direct interaction with the GnRH receptors placed at the surface of ovarian cancer cells [73].

Bombesin-like peptides were initially discovered from frog skin, where they are secreted from cutaneous glands as a means of communication and defense. They were later found to be widely distributed in mammalian neural and endocrine cells represented by the neuromedin B (NMB) and the gastrin-releasing peptide (GRP), respectively. Bombesin-like peptide receptors are G-protein-coupled and have seven membrane-spanning domains, so they are involved in signal transduction pathways [74]. Growing evidence shows that bombesin-like peptides and receptors play essential roles in physiological conditions and diseases. An abnormal expression of bombesin receptors has been observed in several types of tumors, which has motivated the development of more specific and safer bombesin derivatives for tumor diagnosis and therapy [75].

The motif search by using different approaches may render a diversity of outcomes. However, some hits shared by different search approaches can support the reliability of the findings. For example, one motif found by the PROSITE search, WSY, was also encountered by STREME, an algorithm that works regardless of database and sequence similarity. Some of the motifs estimated by MSA algorithms were also identified by the AF approach STREME, such as WWW and WKG. All motifs were searched against TH databases, TumorHoPe, and starPepDB to discriminate the possible new signatures from the existing ones. New motifs appear at very low frequency within THPs (last column of Table 6–8), except CNG found by STREME, which appears 34 times in TumorHoPe and 32 in starPepDB. However, CNG has not been reported as a TH motif.

## 4. Conclusions

In this study, a novel methodology based on network science and similarity searching was introduced to explore the chemical space of THPs and discover potential THPs from known AMPs. Statistically, the strategy’s performance transcended current supervised ML approaches used in THP predictions, demonstrating the potential of this approach. Hence, in silico predictions using the model based on representative THPs, in conjunction with TumorHPD and THPep servers, gave a high reliability to discover potential THPs. As a result, 54 lead compounds were repurposed as potential from AMPs. In the set, novel motifs with promising tumor-homing activity were proposed. 

The good performance of the methodology for predicting peptide activity based on similarity searching and network science suggests its application for the prediction of other endpoints in peptides, e.g., antibacterial activity, toxicity, hemolytic, or anticancer. Our models and pipeline are freely available through the starPep toolbox software at http://mobiosd-hub.com/starpep (accessed on 2 February 2021).

## Figures and Tables

**Figure 1 antibiotics-11-00401-f001:**
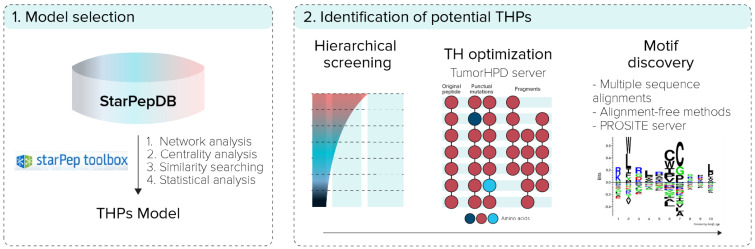
General overview of the experimental procedure.

**Figure 2 antibiotics-11-00401-f002:**
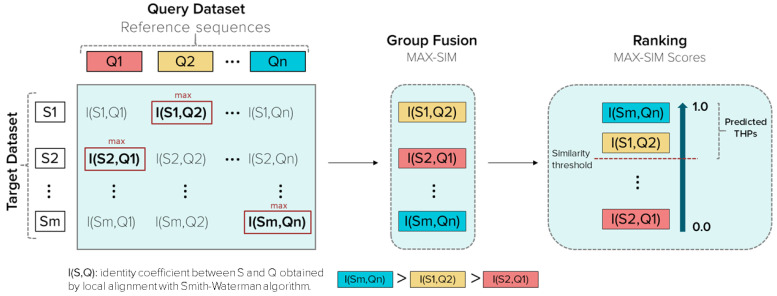
Schematic representation of the group fusion and similarity searching processes. Q is a peptide from a query dataset, n is the number of peptides contained in a query dataset, S is a peptide from the target dataset (Main, Small, or Main90 dataset), m is the number of peptides included in the target dataset (1302, 938, or 619, respectively). The similarity threshold is related to the percentage of identity.

**Figure 3 antibiotics-11-00401-f003:**
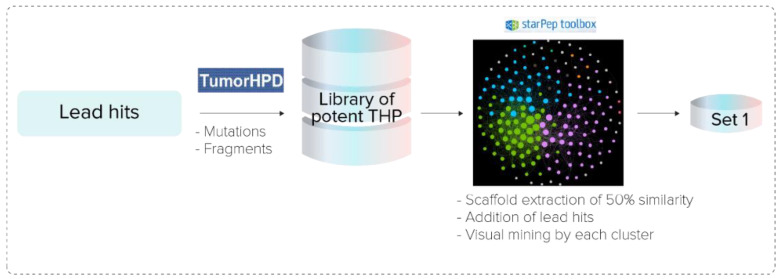
Procedure to optimize tumor-homing activity of lead hits.

**Figure 4 antibiotics-11-00401-f004:**
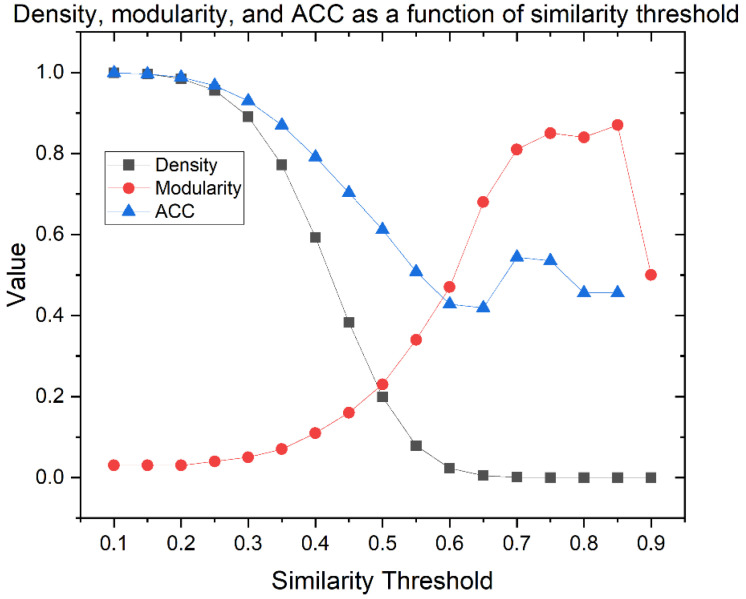
Density, modularity, and average clustering coefficient (ACC) as a function of similarity threshold of 627 THPs CSN.

**Figure 5 antibiotics-11-00401-f005:**
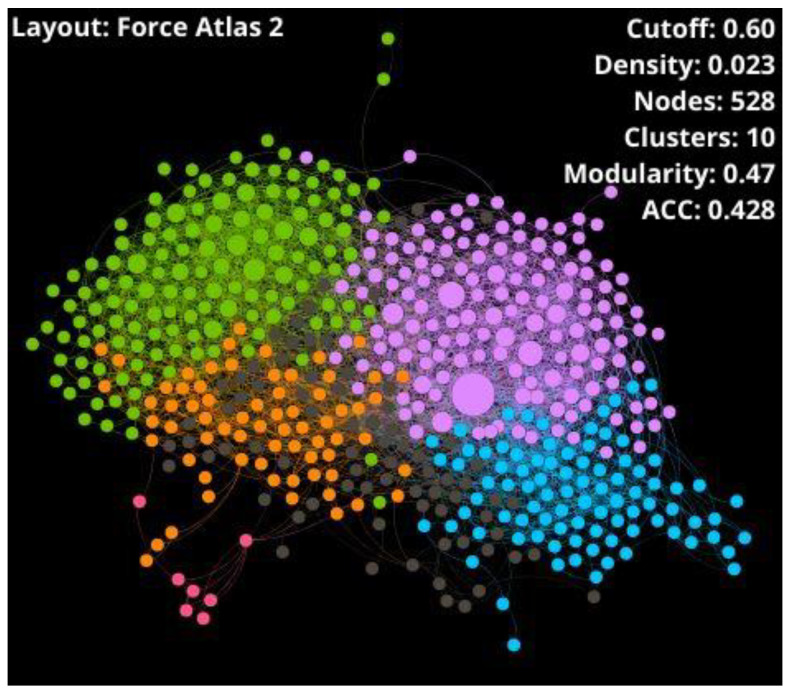
CSN of giant component conformed by 528 THPs retrieved from starPepDB. Node color represents the community (cluster), and node size symbolizes the centrality values.

**Figure 6 antibiotics-11-00401-f006:**
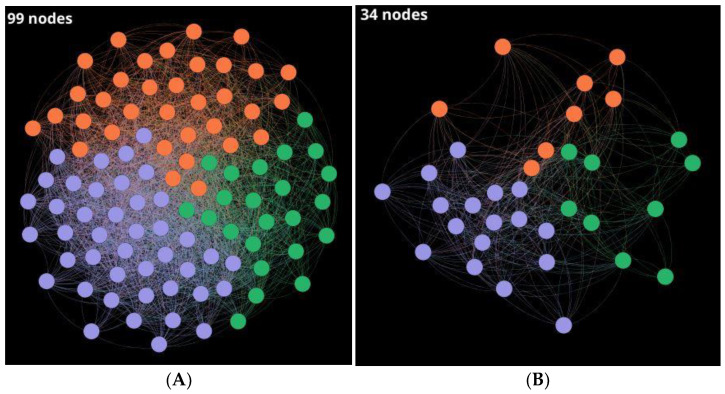
CSN of (**A**) 99 outliers with a density of 0.30 and (**B**) 34 remaining outliers resulting from 30% similarity extraction scaffold. *Layout: Fruchterman–Reingold*.

**Figure 7 antibiotics-11-00401-f007:**
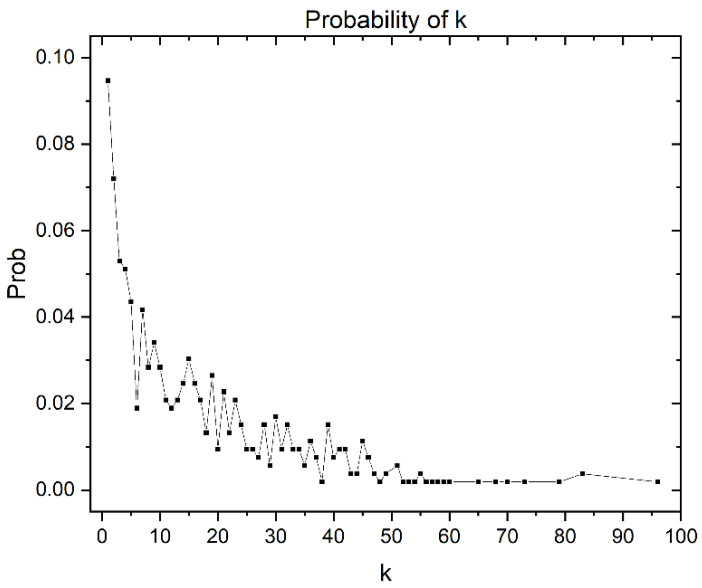
Degree distribution of the 528 giant components, where k is the vertex degree.

**Figure 8 antibiotics-11-00401-f008:**
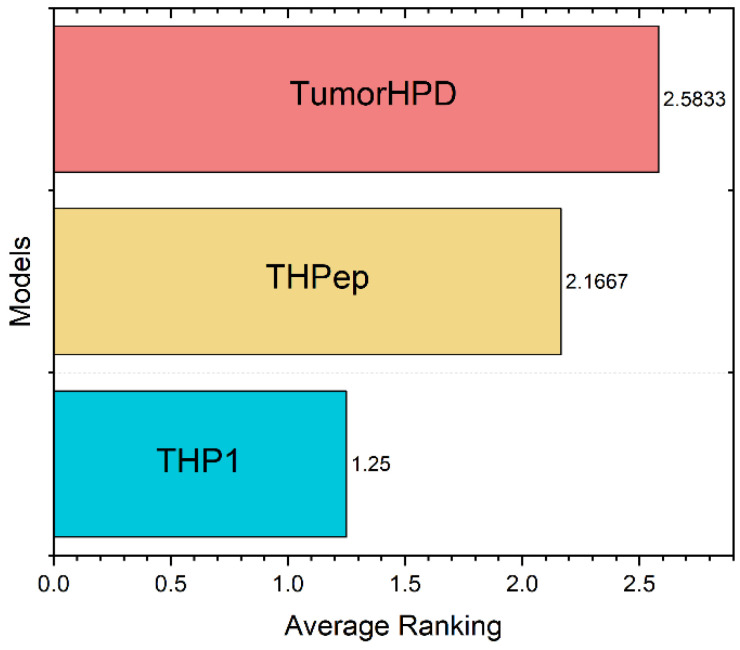
Ranking scores obtained in the Friedman Test. Friedman statistic (distributed according to chi-square with 2 degrees of freedom): 11.166667. P-value computed by Friedman Test: 0.00376.

**Figure 9 antibiotics-11-00401-f009:**
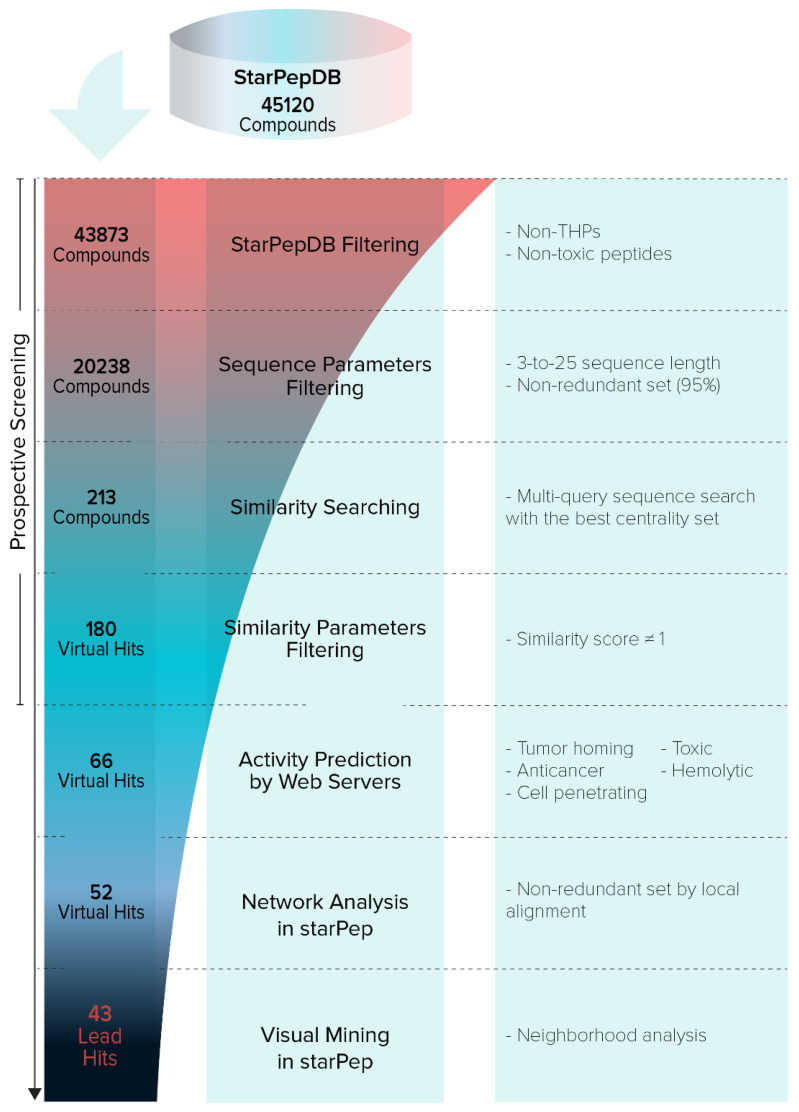
Hierarchical virtual screening for repurposing of peptides from starPepDB.

**Figure 10 antibiotics-11-00401-f010:**
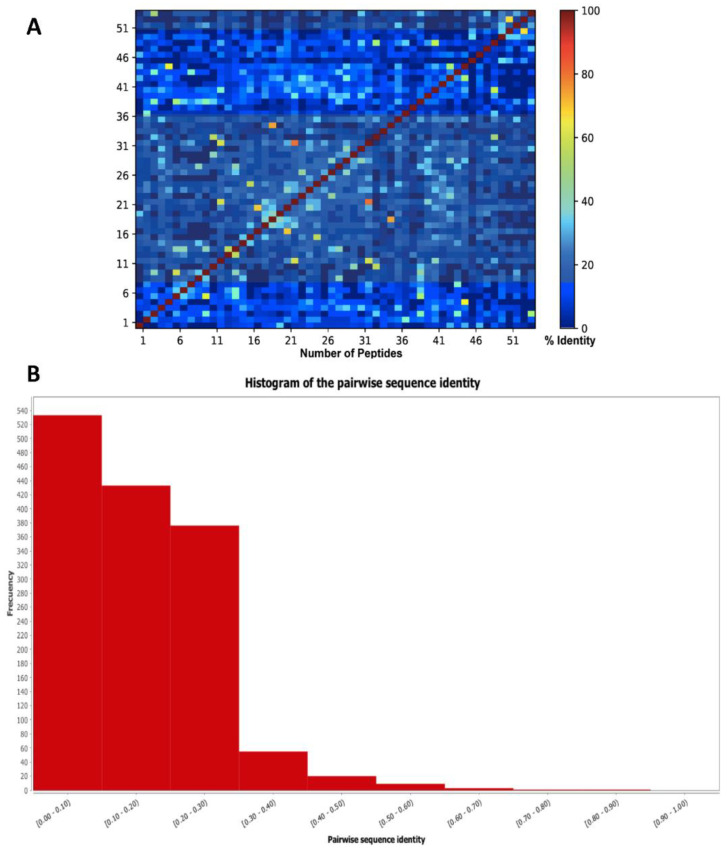
(**A**) Heat map and (**B**) histogram of pairwise sequence identity of SET 1 (54 lead compounds).

**Table 1 antibiotics-11-00401-t001:** Global network properties of CSN of 528 nodes and outliers.

Set *	Nodes	Edges	Density	Clusters	Modularity	Average Degree	ACC	Diameter	Nodes after Sc. **	Edges after Sc. **
THPs	528	4452	0.023	10	0.47	16.864	0.428	8	-	-
Outliers	99	2691	0.891	3	0.13	54.364	0.733	3	34	384

* Density, number of clusters, and modularity were calculated in the starPep toolbox, while average degree, ACC, and diameter were calculated in Gephi. ** Sc.: Scaffold extraction.

**Table 2 antibiotics-11-00401-t002:** Statistical analysis for the performance of the best 9 SSMs on the target Main dataset.

Query Set *	Nodes	% Id	Ac	Correct Class	Incorrect Class	κ	Sn	Sp	P_pos_	P_neg_
**H + sing**	467	40	0.933	1215	87	0.866	0.877	0.989	0.988	0.89
		50	0.935	1218	84	0.871	0.877	0.994	0.993	0.89
		60	0.935	1218	84	0.871	0.874	0.997	0.996	0.888
**W + sing**	469	40	0.934	1216	86	0.868	0.879	0.989	0.988	0.891
		50	0.936	1219	83	0.873	0.879	0.994	0.993	0.891
		60	0.937	1220	82	0.874	0.877	0.997	0.997	0.89
**H + W + sing**	479	40	0.942	1226	76	0.883	0.894	0.989	0.988	0.903
		50	0.944	1229	73	0.888	0.894	0.994	0.993	0.904
		60	0.945	1230	72	0.889	0.892	0.997	0.997	0.903

* H is the set obtained when harmonic centrality was calculated, W is the set obtained when the weighted degree was calculated, and sing is the set of 99 outliers.

**Table 3 antibiotics-11-00401-t003:** Statistical analysis for the performance of the best 9 SSMs on the target Small dataset.

Query Set *	Nodes	% Id	Ac	Correct Class	Incorrect Class	κ	Sn	Sp	P_pos_	P_neg_
**H + sing**	467	40	0.917	860	78	0.834	0.838	0.996	0.995	0.86
		50	0.916	859	79	0.832	0.836	0.996	0.995	0.858
		60	0.914	857	81	0.827	0.832	0.996	0.995	0.855
**W + sing**	469	40	0.92	863	75	0.84	0.844	0.996	0.995	0.865
		50	0.92	863	75	0.84	0.844	0.996	0.995	0.865
		60	0.919	862	76	0.838	0.842	0.996	0.995	0.863
**H + W + sing**	479	40	0.928	870	68	0.855	0.859	0.996	0.995	0.876
		50	0.928	870	68	0.855	0.859	0.996	0.995	0.876
		60	0.926	869	69	0.853	0.857	0.996	0.995	0.875

* H is the set obtained when harmonic centrality was calculated, W is the set obtained when the weighted degree was calculated, and sing is the set of 99 outliers.

**Table 4 antibiotics-11-00401-t004:** Statistical analysis for the performance of the best 9 SSMs on the target Main90 dataset.

Query Set *	Nodes	% Id	Ac	Correct Class	Incorrect Class	κ	Sn	Sp	P_pos_	P_neg_
**H + sing**	467	40	0.985	600	9	0.964	0.983	0.986	0.966	0.993
		50	0.99	603	6	0.976	0.983	0.993	0.983	0.993
		60	0.992	604	5	0.98	0.983	0.995	0.989	0.993
**W + sing**	469	40	0.98	597	12	0.952	0.966	0.986	0.966	0.986
		50	0.984	599	10	0.96	0.966	0.991	0.977	0.986
		60	0.987	601	8	0.968	0.966	0.995	0.988	0.986
**H + W + sing**	479	40	0.985	600	9	0.964	0.983	0.986	0.966	0.993
		50	0.989	602	7	0.972	0.983	0.991	0.977	0.993
		60	0.992	604	5	0.98	0.983	0.995	0.989	0.993

* H is the set obtained when harmonic centrality was calculated, W is the set obtained when the weighted degree was calculated, and sing is the set of 99 outliers.

**Table 5 antibiotics-11-00401-t005:** Comparison between the best SSM THP1 and the state-of-the-art ML model to predict tumor-homing activity of benchmarking datasets.

Dataset	Method	Ac (%)	Sn (%)	Sp (%)	MCC
**Main**	**TumorHPD**	86.56	80.63	89.71	0.7
	**THPep**	86.1	87.07	85.18	0.72
	* **THP1** *	* **94.47** *	* **89.25** *	* **99.66** *	* **0.894** *
**Small**	**TumorHPD**	81.88	73.13	90.92	0.65
	**THPep**	83.37	81.24	85.81	0.67
	*THP1*	*92.64*	*85.71*	*99.5*	*0.861*
**Main90**	**TumorHPD**	89.66	83.64	80.68	0.74
	**THPep**	90.8	91.8	87.97	0.77
	*THP1*	*99.18*	*98.3*	*99.54*	*0.98*

**Table 6 antibiotics-11-00401-t006:** Discovered motifs by Multiple Sequence Alignment (MSA).

No	Motif	EMBOSS Consensus	Cluster	Cluster Size	Frequency *	MSA Method
**1**	wwW	wwW	2	14	1/(1)	CLUSTALW-O
		xxW				MAFFT
**2**	C[fl][rg][vl]rW	CxxxrW	3	10	0/(0)	MAFFT
**3**	C[gpi][gs]cR	CxxxR				MUSCLE
**4**	[rkl]GLC	RGlc	4	8	0/(0)	CLUSTALW-O
		kGLC				MAFFT
		xGLc				MUSCLE
**5**	c[wp]kG	cwkG	1+5	4	0/(0) 0/(0) 0/(1)	CLUSTALW-O MUSCLE
		cxkG				T-Coffee
**6**	Not Found	Non-consensus	6	10	0/(0)	CLUSTALW-O MUSCLE MAFFT T-Coffee
**7**	l[rp][cw]c	lxxc	Singletons	8	0/(0)	MUSCLE

* Taken from TumorHoPe (outside parenthesis), and starPepDB (inside parenthesis).

**Table 7 antibiotics-11-00401-t007:** Discovered Motifs by STREME.

No	Motif	Cluster	Cluster Size	Matches in Positive Seqs.	Matches in Control Seqs.	Sites (%)	Score	Frequency *
**1**	WRP	2	14	7	1	50	1.6 × 10^−2^	5/(5)
**2**	WVL			5	1	35.7	8.2 × 10^−2^	0/(0)
**3**	WS[YR]			3	0	21.4	1.1 × 10^−1^	1/(1)Y
**4**	WWWM			3	0	21.4	1.1 × 10^−1^	0/(0)
**5**	CFRV	3	10	3	0	30	1.1 × 10^−1^	1/(1)
**6**	HWK			2	0	20	2.4 × 10^−1^	0/(0)
**7**	PRW			2	0	20	2.4 × 10^−1^	3/(3)
**8**	CN[WG]	4	8	3	0	37.5	1.0 × 10^−1^	34/(32)G
**9**	WARG			3	0	37.5	1.0 × 10^−1^	0/(0)
**10**	GIC			2	0	25.0	2.3 × 10^−1^	5/(4)
**11**	WKG	1-5	4	3	1	75.0	2.4 × 10^−1^	0/(0)
**12**	KNKHK	6	10	3	0	30.0	1.1 × 10^−1^	0/(0)
**13**	PSHL			3	0	30.0	1.1 × 10^−1^	0/(0)
**14**	LRLRI	Singletons	8	2	0	25.0	2.3 × 10^−1^	1/(1)
**15**	CC[CQ]			3	1	37.5	2.8 × 10^−1^	0/(0)
**16**	LSP	All sequences	54	11	1	20.4	3.4 × 10^−3^	3/(3)
**17**	WSYG			7	0	13.0	8.2 × 10^−3^	0/(0)
**18**	WRPW			5	0	9.3	3.2 × 10^−2^	2/(2)

* Taken from TumorHoPe (outside parenthesis), and starPepDB (inside parenthesis).

**Table 8 antibiotics-11-00401-t008:** Motifs found in PROSITE.

No	Motif Found	Hit Peptide	Accession	Match with	Signature	Related Seqs.	Frequency *
**1**	QHWSYGLRPG	starPep_07237	PS00473	Q[HY][FYW]Sx(4)PG	Gonadotropin-releasing hormones	67	1/(1)QHWSY
**2**	WARGHFM	starPep_10020	PS00257	WAxG[SH][LF]M	Bombesin-like peptides	36	0/(0)

* Taken from TumorHoPe (outside parenthesis), and starPepDB (inside parenthesis).

## Data Availability

The starPep toolbox software and the respective user manual, as well as *SSMs*, are freely available online at http://mobiosd-hub.com/starpep (accessed on 2 February 2021).

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
