# Peer review of "A Novel Network Science and Similarity-Searching-Based Approach for Discovering Potential Tumor-Homing Peptides from Antimicrobials"

_antibiotics, 2022, doi:10.3390/antibiotics11030401_

Round 1

Reviewer 1 Report

Romero and co-workers presented a network science and similarity search based approach to discover potential THPs from anti-microbial peptides. The manuscript was well written with a good introduction to THPs and various databases currently available to validate THPs. The approach and results were clearly presented.

The manuscript has some grammatical errors, and minor spelling mistakes. After correction it is ready for publication  

Author Response

R1: Romero and co-workers presented a network science and similarity search based approach to discover potential THPs from anti-microbial peptides. The manuscript was well written with a good introduction to THPs and various databases currently available to validate THPs. The approach and results were clearly presented.

The manuscript has some grammatical errors, and minor spelling mistakes. After correction it is ready for publication  

Answers

Thank you very much for your time in reviewing the article and for your opinion. The manuscript was revised and the English improved

Reviewer 2 Report

In this article, the authors developed a new methodology based on network science and similarity searching to explore the chemical space of THPs and discover potential THPs from known AMPs. The performance of the strategy transcends current supervised ML approaches used in THPs predictions, demonstrating its great potential in the prediction of not only THPs but also other endpoints in peptides, such as antibacterial activity, toxicity, hemolytic, and anticancer. Moreover, 54 peptides with diverse molecular structures are predicted to exhibit low toxicity, hemolytic activity, and potential anticancer activity, which may provide guidelines for the discovery of new THPs.

Therefore, I would like to recommend its publication in Antibiotics after the following minor concerns are addressed.

1. Page 12, Line 468. There are some format issues when citing Table 2, Table 3, and Table 4. Same as Page 14, Line 492.

2. The resolution of Figure 10 needs to be improved.

3. Page 21, Line 644. “the performance of the strategy’s performance” should be “the strategy’s performance”.

4. Ref 42 is not properly cited. Please double-check.

Author Response

Referee 2

In this article, the authors developed a new methodology based on network science and similarity searching to explore the chemical space of THPs and discover potential THPs from known AMPs. The performance of the strategy transcends current supervised ML approaches used in THPs predictions, demonstrating its great potential in the prediction of not only THPs but also other endpoints in peptides, such as antibacterial activity, toxicity, hemolytic, and anticancer. Moreover, 54 peptides with diverse molecular structures are predicted to exhibit low toxicity, hemolytic activity, and potential anticancer activity, which may provide guidelines for the discovery of new THPs.

Therefore, I would like to recommend its publication in Antibiotics after the following minor concerns are addressed.

R1: 1. Page 12, Line 468. There are some format issues when citing Table 2, Table 3, and Table 4. Same as Page 14, Line 492.

Answers A1

  1. Line 98

“high similarity sequences” changed to “highly similar sequences”.

  1. Figures 4, 7 and 8

The decimal marks were changed to dots.

  1. Line 467

“The performance of the best nine SSMs to predict activity in Main, Small and Main90 datasets are shown in Table 2, *H is the set obtained when harmonic centrality was calculated, W is the set obtained when the weighted degree was calculated, and sing is the set of 99 outliers.

Table 3 and *H is the set obtained when harmonic centrality was calculated, W is the set obtained when the weighted degree was calculated, and sing is the set of 99 outliers.

Table 4, respectively.” changed to “The performance of the best nine SSMs to predict activity in Main, Small and Main90 datasets are shown in Table 2, Table 3 and Table 4, respectively, where H is the set obtained when harmonic centrality was calculated, W is the set obtained when the weighted degree was calculated, and sing is the set of 99 outliers”.

  1. Line 494

“Table 2, *H is the set obtained when harmonic centrality was calculated, W is the set obtained when the weighted degree was calculated, and sing is the set of 99 outliers.

Table 3, and *H is the set obtained when harmonic centrality was calculated, W is the set obtained when the weighted degree was calculated, and sing is the set of 99 outliers.

Table 4).” changed to “(shown in Table 2, Table 3 and Table 4)”.

R2: The resolution of Figure 10 needs to be improved.

Answers A2

The resolution was improved and update inside of main text. Thanks!

Answers A2

Page 21, Line 644. “the performance of the strategy’s performance” should be “the strategy’s performance”.

Ok, thanks!

R4: Ref 42 is not properly cited. Please double-check.

Answers A4

Ref 42

“Shoombuatong, W.; Schaduangrat, N.; Nantasenamat, C. UNRAVELING THE BIOACTIVITY OF ANTICANCER PEPTIDES AS. EXCLI J. 2018, 17, 734–752, doi:10.17179/excli2018-1447 This.” changed to “Shoombuatong, W.; Schaduangrat, N.; Nantasenamat, C. Unraveling the bioactivity of anticancer peptides as deduced from machine learning. EXCLI J. 2018, 17, 734–752, doi:10.17179/excli2018-1447.”

Reviewer 3 Report

The authors present a novel methodology based on network science and similarity searching to explore the chemical space of tumor homing peptides (THPs) with tha aim to find new potential THPs from antimicrobial peptides. I general, the work is well done and will be of interest to those in peptide-based drug research area. Just minor  details before its acceptance for publication.

1.- Please improve the quality of figure 10, is difficult to read it.

2.- The wording is confusing at the beginning of lines 471, 473, 495, and 497.

Author Response

Referee 3

The authors present a novel methodology based on network science and similarity searching to explore the chemical space of tumor homing peptides (THPs) with the aim to find new potential THPs from antimicrobial peptides. I general, the work is well done and will be of interest to those in peptide-based drug research area. Just minor details before its acceptance for publication.

R1: Please improve the quality of figure 10, is difficult to read it.

Answers A1

This figure was update with their quality was improved, thanks!

R2: The wording is confusing at the beginning of lines 471, 473, 495, and 497.

Answers A2

lines 471, 473…OK, the wording was adjusted to improve the idea and avoid confusion, Thanks!!!

The performance of the best nine SSMs to predict activity in Main, Small and Main90 datasets are shown in Table 2, Table 3 and Table 4, respectively, where H is the set obtained when harmonic centrality was calculated, W is the set obtained when the weighted degree was calculated, and sing is the set of 99 outliers

lines 495, 497…OK, …OK, the wording was adjusted to improve the idea and avoid confusion, Thanks!!!

The best nine SSMs were compared and ranked using the Friedman test by comparing multiple statistical metrics from each SSM on the three target datasets (details in SI6-B). The best SSM was the set CSN-TH-0.60Sc-479-H+W+s-0.6-583 (479Q_0.6), named THP1, showing excellent statistical metrics (>0.85) for the model (shown in Table 2, Table 3 and Table 4).